# Antibody Profiling and In Silico Functional Analysis of Differentially Reactive Antibody Signatures of Glioblastomas and Meningiomas

**DOI:** 10.3390/ijms24021411

**Published:** 2023-01-11

**Authors:** Lisa Milchram, Ronald Kulovics, Markus Sonntagbauer, Silvia Schönthaler, Klemens Vierlinger, Christian Dorfer, Charles Cameron, Okay Saydam, Andreas Weinhäusel

**Affiliations:** 1Center for Health and Bioresources, Competence Unit Molecular Diagnostics, AIT Austrian Institute of Technology GmbH, Giefinggasse 4, 1210 Vienna, Austria; 2Department of Neurosurgery, Medical University of Vienna, 1090 Vienna, Austria; 3Department of Pediatrics, Medical School, University of Minnesota, 420 Delaware Street, Minneapolis, MN 55455, USA

**Keywords:** glioblastoma, meningioma, brain tumors, protein microarrays, autoantibodies, seroreactivity, pathway analysis, tumor-autoantibodies, tumor-antigens

## Abstract

Studies on tumor-associated antigens in brain tumors are sparse. There is scope for enhancing our understanding of molecular pathology, in order to improve on existing forms, and discover new forms, of treatment, which could be particularly relevant to immuno-oncological strategies. To elucidate immunological differences, and to provide another level of biological information, we performed antibody profiling, based on a high-density protein array (containing 8173 human transcripts), using IgG isolated from the sera of *n* = 12 preoperative and *n* = 16 postoperative glioblastomas, *n* = 26 preoperative and *n* = 29 postoperative meningiomas, and *n* = 27 healthy, cancer-free controls. Differentially reactive antigens were compared to gene expression data from an alternate public GBM data set from OncoDB, and were analyzed using the Reactome pathway browser. Protein array analysis identified approximately 350–800 differentially reactive antigens, and revealed different antigen profiles in the glioblastomas and meningiomas, with approximately 20–30%-similar and 10–15%-similar antigens in preoperative and postoperative sera, respectively. Seroreactivity did not correlate with OncoDB-derived gene expression. Antigens in the preoperative glioblastoma sera were enriched for signaling pathways, such as *signaling by Rho-GTPases, COPI-mediated anterograde transport and vesicle-mediated transport,* while the *infectious disease, SRP-dependent membrane targeting cotranslational proteins* were enriched in the meningiomas. The pre-vs. postoperative seroreactivity in the glioblastomas was enriched for antigens, e.g., *platelet degranulation* and *metabolism of* lipid pathways; in the meningiomas, the antigens were *enriched in infectious diseases, metabolism of amino acids and derivatives,* and cell cycle. Antibody profiling in both tumor entities elucidated several hundred antigens and characteristic signaling pathways that may provide new insights into molecular pathology and may be of interest for the development of new treatment strategies.

## 1. Introduction

Glioblastomas are the most aggressive type of brain tumor with poor prognosis, with an incidence of 3.19–4.17 cases per 100,000 person-years [1]. Meningiomas, on the other hand, are among the most common tumors of the central nervous system, and they originate in arachnoid cells [2]. The standard diagnostics for both tumors are imaging methods (CT, MRI). The formation of tumor-associated antibodies (TAAs)—more precisely, the induction of antibody production by B cells— during the transformation and progression of cancer cells, has been known of for several decades. TAAs are considered to be potential biomarkers of early cancer, due to their excellent stability and prior formation of clinical tumor manifestation [3,4,5]. In addition to their biomarker potential, several studies have promoted the potential of TAAs to elucidate biological signatures. These antibody profiles are an additional layer of omics, that can also be *called the immunome*, due to their broad spectrum of reactive binding of antibodies, elucidated in parallel on protein arrays. In previous work, we have shown that the immunome provides biologically meaningful information in cancer [6,7,8] and autoimmune diseases (rheumatoid arthritis) [9], and in inflammatory diseases (ulcerative colitis, [10]). Regarding the current trend in immuno-oncology, antigenic reactivity has aroused great interest, and several therapeutic approaches are being considered and studied, to improve glioblastoma treatment.

While glial-derived glioblastoma is the most serious form of malignant brain tumor, meningiomas are generally slow-growing benign tumors that arise from the arachnoid cap cells of the leptomeninges, the soft coverings of the brain and spinal cord. The current state of knowledge on characteristics and therapy options was recently reviewed [11,12]. While the preferred treatment option for meningiomas is, in many cases, observation, the standard therapy—if needed—is surgery, with or without adjuvant radiation, depending on the tumor grade and the degree of resection. Systemic therapies are not, as yet, part of standard care. The evidence base for treatment recommendations is scarce, and effective treatment regimens, particularly for treatment-refractory and radiation-refractory meningiomas, are still very limited [12].

Although glioblastomas and meningiomas are vastly different, the discovery of the genetic and epigenetic signatures has provided insight into their biology, and has enabled the identification of novel biomarker candidates and potential therapeutic targets for glioblastomas and refractory meningiomas. We present here our immunological approach to elucidating another layer of biological information, which employed antibody profiling based on a high-density protein array, using IgG purified from sera of preoperative and postoperative glioblastomas and meningiomas, compared to cancer-free controls, in order to (I) identify differentially reactive antigens (DIRAGs) for both tumor groups, which (II) were also analyzed using pathway analysis, to gain insights into the underlying biological processes, as shown in Figure 1.

## 2. Results

To identify and study tumor-associated antibody signatures in glioblastomas and meningiomas, serum IgG was probed on high-density protein microarrays. IgG was isolated from pooled preoperative and postoperative sera from patients with glioblastomas (GBM, *n* = 12 preoperative, and *n* = 16 postoperative) and from patients with meningiomas (MEN, *n* = 26 preoperative, and *n* = 29 postoperative), and from 27 cancer-free controls, which were probed on AIT’s 16 k protein microarray that included human proteins with *n* = 6124 annotated human genes and 8173 different annotated human transcripts, recombinantly expressed from 15,312 cDNA *E. coli* clones. Fluorescence data obtained from microarray images were analyzed for differentially reactive antigens (DIRAGs) between preoperative and postoperative glioblastomas and meningiomas, and were compared to cancer-free controls. Higher reactive proteins within the case groups were subjected to the Reactome pathway browser, to examine the underlying disease pathways and altered pathways in pre-surgery vs. controls and in pre-surgery-vs.-post-surgery patients. In addition, antibody reactivity data in glioblastomas were compared to public gene expression data from OncoDB, as summarized in Figure 1.

### 2.1. IgG Concentration in GBM and MEN Sera

For protein array processing, we purified IgG, and analyzed concentration-standardized IgG amounts on protein arrays: this gave us the opportunity to also test for differences in serum IgG concentrations, in relation to the different sample groups, when measuring IgG concentrations during serum purification.

The concentration of IgG isolated from plasma was determined by absorbance at 280 nm (A280) in duplicate. The median IgG concentration obtained from all 144 samples was 6.70 ± 3.54 mg/mL serum. The group mean values of the calculated serum IgG concentration were as follows: preoperative glioblastomas, 6.78 ± 2.19 mg/mL; preoperative meningiomas, 6.70 ± 2.37 mg/mL; postoperative glioblastomas, 6.76 ± 1.12 mg/mL; postoperative meningiomas, 5.85 ± 1.48 mg/mL; and cancer-free controls, 7.99 ± 4.95 mg/mL.

We applied the estimation statistics web tool (https://www.estimationstats.com/ accessed on 6 August 2022) to analyze potential differences in serum IgG concentrations, using the median difference for four comparisons with the common healthy controls [15]. The unpaired median difference between control and GBM-pre, GBM-post and MEN-pre were not significant, whereas the difference between control and MEN-post was significantly reduced, by 2.13 mg/mL (*p* = 0.0128; two-sided permutation t-test), as shown in Figure 2. Pairwise comparison of IgG concentration in sera before and after surgery did not reveal a significant reduction (median −0.118 mg/mL) in IgG concentration in postoperative GBM patients. We observed a statistically significant reduction in IgG concentration in postoperative MEN patients (a median (−1.2 mg/mL) dp = 0.0176; two-tailed permutation *t*-test), as shown in Figure 2 (right). We broke 34 arrays while processing, leaving only 110 samples to be analyzed. For 110 of these 144 samples, antibody profiles derived from 16 k protein array data (as shown in Figure 1) were available for analysis.

### 2.2. Antibody Profiling on 16 k Protein Arrays

The following preprocessed and COMBAT and quantile-normalized [16] IgG profile data from protein arrays underwent bioinformatic analysis, to identify significant (*p* < 0.05) differentially reactive antigens (DIRAGs) for respective class comparison: (I) pre/postoperative glioblastomas (GBM-pre/GBM-post) and pre-/postoperative meningiomas (MEN-pre/MEN-post) vs. healthy control—see contrasts (a–d) in Figure 3; (II) preoperative vs. postoperative GBM and MEN—see contrasts (e) and (f) in Figure 3. Of *n* = 8254 remaining features after preprocessing and filtering, exploratory analysis of tumors vs. controls, for the contrasts (a) *n* = 246, (b) *n* = 308, (c) *n* = 375, and (d) *n* = 199, revealed significantly higher reactive antigens in GBM-pre, GBM-post, MEN-pre, and MEN-post, respectively. In contrast (e), *n* = 151 antigens were more highly reactive in GBM-pre, and *n* = 155 were more highly reactive in GBM-post; in (f) *n* = 232, the antigens were more highly reactive in MEN-pre, and *n* = 206 were more highly reactive in MEN-post.

Comparing the number of DIRAGs from (a) GBM-pre vs. controls vs. (c) MEN-pre vs. controls, in (a) 246 and in (c) 375 DIRAGs were higher reactive in cases, whereas in both contrasts the number of higher reactive DIRAGs in controls yielded very similar numbers, of 307 and 311 DIRAGS. Comparing the contrasts (a) GBM-pre vs. controls and (b) GBM-post vs. controls, an increased number of 308 DIRAGs was higher reactive in GBM-post than the 246 DIRAGS in (a); in MEN vs. controls, contrasts (c) and (d), antigenic reactivity was lower in MEN-post, with 199 DIRAGS in (d), compared to 375 in MEN-pre, of contrast (c). The comparison of the number of pre- and post-operations in GBM—contrast (e)—resulted in about 150 DIRAGS in both directions. The numbers of DIRAGS in GBM were lower than in meningioma, where we found higher reactive DIRAGS in MEN-pre, *n* = 206, and higher in MEN-post, *n* = 232—contrast (f). Visual comparison of DIRAGS, presented in volcano plots illustrating the mean antigenic reactivities of individual antigens tested in the compared sample groups, shows slightly higher fold changes in GBM contrasts (a,b,e) than in MEN contrasts (c,d,f) which have lower fold-changes in antigenic reactivities (Figure 3B). The complete tables of DIRAGS from class comparisons are provided in the Appendix A.

The top 10 differentially reactive proteins, based on the mean change in reactivity between classes of these antigens, are summarized in the Table 1 glioblastoma contrasts (a,c,e) and in the Table 1 meningioma contrasts (c,d,f). The *p*-values and fold changes derived from the protein array analysis were in a moderate range, which was consistent with our previous studies performed on other cancers, and also on inflammatory (ulcerative colitis) and autoimmune (rheumatoid arthritis) diseases [9,10]. The diagnostic potential of serotesting showed the potential of the candidate biomarkers based on the AUC values, although the sample numbers were low, and need to be confirmed in larger studies. For a potential initial diagnosis according to contrast (a), GBM-pre vs. control shows AUC values of a single antigen in a range of AUC 0.7–0.8, suggesting potential biomarkers for glioblastomas. The significant DIRAGS (based on unique gene symbols) derived from different contrasts are compiled and shown in Figure 3A, and overlapping and non-overlapping of the genes in glioblastomas and meningiomas are shown in the respective Venn diagrams (Figure 3B), as described.

### 2.3. Overlap of DIRAGs

The preoperative IgG reactivity of the glioblastomas showed *n* = 246 significantly higher reactive DIRAGs compared to healthy controls (Figure 3A(a)); postoperatively, *n* = 308 significantly higher reactive DIRAGs were detected (Figure 3A(b)). The comparison of preoperative and postoperative glioblastomas showed *n* = 151 significant DIRAGs in preoperative glioblastomas, and *n* = 155 significant DIRAGs in postoperative glioblastomas (Figure 3A(e)). Overlaps between the contrasts are shown in Figure 3B. Of 246 DIRAGs higher reactive in GBMs before surgery (vs. healthy controls), 59 (24%) overlapped DIRAGs that were higher reactive in GBMs after surgery (vs. healthy controls); of the same 246 DIRAGs, 45 (20%; red and green ellipses in Figure 3B—left) retained higher reactivity in GBMs before surgery compared to GBMs after surgery (derived from contrast (e)). Similarly, in contrast (b), of 308 DIRAGs higher reactive in post-surgery GBMs vs. controls, *n* = 58 (19%) were found also higher reactive in post-surgery GBMs compared to pre-surgery GBMs (contrast (e)). The DIRAGs higher reactive in GBM-pre and GBM-post vs. controls, from contrasts (a) and (b), as well as those from contrast (e), from both higher and less reactive in either group, and the intersection of these protein lists, are provided in Appendix A.

Meningioma serum-IgG before surgery showed *n* = 375 significantly higher reactive DIRAGs (Figure 3B; contrast (c)), and postoperative, *n* = 199 significantly higher reactive DIRAGs, compared to healthy controls. The contrast (f) of preoperative vs. postoperative meningiomas showed *n* = 206 significantly higher DIRAGs in preoperative meningiomas, and *n* = 232 in postoperative meningiomas. Of 375 DIRAGs with higher reactivity in MEN-pre vs. healthy (contrast (c)), and 232 with higher reactivity in MEN-pre compared to MEN-post (contrast (f)), *n* = 101 (101/375 = 27%) overlapped. Thirty-five (*n* = 35)—18% of the more reactive DIRAGs (*n* = 199)—in postoperative meningiomas compared to controls (contrast (d)) were also more reactive compared to preoperative meningiomas (206 higher in MEN-post vs. (f)). A comparison of the antigens that were more reactive in the case groups vs. controls, as derived from preoperative contrasts (a) and (c) vs. postoperative contrasts (b) and (d), revealed an overlap of 59 DIRAGs in (a) and (b) for GBM, and a comparable number of 56 overlapping DIRAGs in (c) and (d) for MEN. Comparison of overlapping antigens from GBM contrasts (a) and (e), *n* = 49 and *n* = 13, and (b) and (e), *n* = 49 and *n* = 13, with numbers from MEN contrasts (c) and (f), *n* = 101 and *n* = 12, and (d) and (f), *n* = 8 and *n* = 35, show comparable dynamics in numbers, but a higher number (*n* = 101) of overlapping DIRAGs in (c) and (f1) (see Appendix A).

When comparing the higher reactive GBM-pre and MEN-pre antigens from contrasts (a) and (c), we found an overlap of 73 antigens (73/246 ≈ 30%; 73/375 = 20%). Comparing the antigens from post-operative contrasts (b) and (d), which showed higher reactivity in GBM-post and in MEN-post, we found an overlap of 30 antigens (30/308 ≈ 10%; 30/199 ≈ 15%; Appendix A). Thus, protein array analysis shows different antigenic profiles in GBM and MEN, with approximately 20%- and 30%-similar, and 10%- and 15%-similar antigens in preoperative and postoperative sera, respectively.

The lists of antigens and intersections of these protein lists are given in Appendix A.

### 2.4. Pathway Analysis of DIRAGs

Reactome pathway analysis was performed, using significantly higher reactive antigens in different contrasts (a–f, see table in Figure 3A). In detail, we (I) examined the GBM contrasts, by comparing (1) GBM-pre vs. healthy (a), (2) GBM-post vs. healthy (b), and (3) GBM-post vs. GBM-pre (e), and then compared in a similar way (II) meningiomas contrasted (c,d,f). The 10 most important signaling pathways derived from the Reactome analysis are summarized in Appendix A. The 10 most important signaling pathways identified from the individual analysis showed *p*-values of *p* < 0.0263, with a median of *p* = 0.0046. The adjusted *p*-values ranged from *p* = 0.066–0.799, with a median *p* = 0.470. The results of the pathway analysis are summarized below, and the two most important pathways, based on the number of antigens found, were identified for selected published contexts examining GBM and comparing it with gene expression data from OncoDB.

#### 2.4.1. Pathways Enriched in GBM-Pre vs. Healthy

##### GBM-Pre vs. Healthy and Antigens with Higher Seroreactivity in GBM-Pre

In contrast (a)—GBM-pre vs. healthy, proteins showing higher seroreactivity in GBM-pre were enriched in pathways signaling by Rho GTPases, COPI-mediated anterograde transport, RHO(D) GTPase cycle, vesicle-mediated transport, and pathways in the context of HIV elongation arrest and recovery. Signaling by Rho GTPases and vesicle-mediated transport pathways had the highest numbers, 29 and 24, of affected genes/antigens, respectively.

It has been shown that in the context of antigenicity, overexpression of Rho-GTPases, or altered expression of proteins associated with these signaling pathways, could lead to increased antigenic IgG reactivity. A very effective search—using ONCODB expression data (http://oncodb.org/download/expression/expression.zip; http://oncodb.org/cgi-bin/genomic_normal_expression_search.cgi accessed on 30 August 2022)—of gene expression in glioblastomas (*n* = 148) vs. normal (*n* = 200), was carried out systematically for the selected top pathways. The search is described below, and also in the following sections, for the pathways found in other contrasts comparing antigenic reactivities deduced from our 16 k protein array analysis. OncoDB was searched for the corresponding gene expression data (using a cut-off of FDR < 0.05), to capture genes overexpressed either in GBM tissue or in normal tissue.

Of the 29 antigens that are higher reactive in GBM-pre, and represent the signaling by the Rho GTPases, Miro GTPases and RHOBTB3 pathways, the retrieved 8 genes (LMNB1, NCKAP1L, ACTG1, PPP1CC, ARHGEF10L, MYO19, CCT6A, TAX1BP3) were significantly overexpressed. On the other hand, 11 of these antigens (ARFGAP2, SPTAN1, KIDINS220, MRTFA, NSFL1C, PLXNA1, MYH10, KIF5A, KLC2, TUBB2A, ITSN1) were found to be significantly higher expressed in normal tissue, using the ONCODB-GBM expression data.

##### GBM-Pre vs. Healthy: Higher Seroreactivity in Healthy

In testing the antigens which were more reactive in healthy sera compared to the GBM pre-sera of contrast (a), COPI-mediated anterograde transport was used in addition to significant signaling pathways, which is also found in the significant pathways identified by antigens with higher reactivity in GBM-pre, but covers different proteins. The **neutrophil degranulation** and **asparagine *n*-linked glycosylation** pathways showed the highest number of antigens—21 and 15, respectively. By comparative analysis with gene expression data from the OncoDB GBM dataset of the 21 antigens that are more reactive in healthy individuals, and are presented in the **neutrophil degranulation** pathway, 13 (ITGAL, DDOST, LAMTOR2, PA2G4, GGH, PSMA2, TUBB, FTL, EEF1A1, PECAM1, DYNLT1, GLB1, GSTP1) were found higher expressed (FDR < 0.05) in GBM, and 5 (DYNC1H1, CD47, VAPA, SPTAN1, ALAD) were found higher expressed (FDR < 0.05) in normal tissue.

Of the 15 antigens that were higher in healthy tissue, and presented in the **asparagine N-linked glycosylation** pathway, TUBA1A, COG4, ALG3, DDOST, GMPPB, GLB1, were found higher expressed (FDR < 0.05) in GBM, and DYNC1H1, DCTN1, TUBB2A, GBF1, MVD, SPTAN1, and ARF1 were found higher expressed (FDR < 0.05) in normal tissue.

#### 2.4.2. Pathways Enriched in GBM-Post vs. Healthy

**The mRNA-splicing major** and **axon guidance pathways** showed the highest number of antigens, which were more highly reactive in GBM-post, as compared to the seroreactivities of healthy individuals (contrast (b)): of 16 antigens found to be higher reactive in GBM-post, and present in the mRNA-splicing major pathway, SART1, PQBP1, RBM17 and POLR2A in healthy tissue, and 8 antigens (U2AF1, EFTUD2, HNRNPH1, HNRNPL, PRPF40A, TRA2B, CTNNBL1 and LSM4) were more strongly expressed in GBM tissue. Of 28 antigens present in the **axon guidance pathway, 9** (DPYSL2, UBB, MYH10, USP33, ANK3, SPTAN1, SH3GL2, ITSN1 and AP2M1) were overexpressed in healthy individuals, and 13 (GPC1, RPL36A, RPS9, RPS2, ROBO3, RPL27, MYL6, RPS15, RPS18, TUBA1A, RPL13, RPS21 and ACTG1) were overexpressed in GBM tissue.

**Hemostasis** and **G2/M transition** showed the most antigens higher reactive in healthy sera, compared to GBM-post. Of 28 **hemostasis** antigens higher reactive in healthy, 10 (CYB5R1, SRI, TUBB6, GNAI2, RAC1, HDAC1, MICAL1, A2M, SLC3A2 and TUBB3) were overexpressed in GBM, and 14 (CALM2, MAPK3, GLG1, PPP2R1A, KIFAP3, PPP2R5D, BRPF3, SH2B1, YWHAZ, SCG3, ACTN2, AAMP, JMJD1C and PPP2R5C) were overexpressed in normal tissue. Of the 17 **G2/M transition** pathway antigens, 5 were overexpressed in the OncoDB GMB tissue samples (TUBB6, RPS27A, RBBP4, TUBB and TUBB3) and 7 (CSNK1D, PPP2R1A, LCMT1, DYNC1H1, PSMD7, NINL and DCTN1) were higher expressed in normal tissue.

#### 2.4.3. Pathways Enriched in GBM-Post vs. GBM-Pre

**Platelet degranulation** and **metabolism of lipids** were the major pathways for antigens, which were higher reactive in GBM-pre vs. GBM-post (contrast (e)), presenting 7 and 15 DIRAGs, respectively. Of the 7 antigens present in the **platelet degranulation** pathway (SCG3, TAGLN2, BRPF3, ALB, CALM2, ACTN2 and MAGED2), two genes (TAGLN2 and MAGED2) were overexpressed in GMB vs. normal tissue; four genes (SCG3, BRPF3, CALM2 and ACTN2) were higher expressed in normal tissue, while ALB was not differentially expressed in the OncoDB dataset. Out of 15 **metabolism of lipid** antigens, 8 (INPPL1, PPP1CA, MED14, DBI, MED21, CPNE1, HMGCL and RUFY1) were overexpressed in GMB, but PLEKHA5 and FASN were overexpressed in normal tissue.

#### 2.4.4. Pathways Enriched in MEN-Pre, MEN-Post, and Healthy

In contrast (c), MEN-pre vs. healthy, in addition to COPI-mediated anterograde transport and N-linked glycosylation pathways also found in the GBM contrasts, disease and signaling by receptor tyrosine kinases pathways were found in 51 and 31 present DIRAGs higher reactive in MEN-pre. Asparagine N-linked glycosylation pathways and metabolism of amino acids and derivatives pathways were those found within the antigens that were higher reactive in healthy subjects, compared to MEN-pre. A direct comparison with expression data was not performed for the MEN contrasts, because no corresponding expression data were available in OncoDB.

Comparison of signaling pathways in contrast (d), MEN-post vs. healthy, showed that COPI-mediated anterograde transport and ER-to-Golgi anterograde transport were the most prominent pathways of DIRAGs higher, which were higher reactive in MEN-post than in controls. COPI-mediated anterograde transport was also demonstrated, in addition to contrast (c), but ER-to-Golgi anterograde transport was found higher reactive in healthy sera, compared to MEN-pre.

Similar to comparing the pathways found to be significant in different contrasts, such as (c) MEN-pre vs. healthy and (d) MEN-post vs. healthy, as above, a direct analysis of the contrast (f) MEN-pre vs. MEN-post placed the infectious disease pathway at the top of the signaling pathways, with 34 DIRAGS identified in MEN-pre as more highly reactive, followed by L13a-mediated translational silencing of ceruloplasmin expression and GTP hydrolysis, and joining of the 60S ribosomal subunit pathways (each presented by 11 antigens). SRP-dependent cotranslational protein targeting to the membrane pathway (10 antigens) has also been found in pathways of contrast (c), similar to the infectious disease pathway. Eukaryotic translation initiation and cap-dependent translation initiation are found on both sides of higher reactive antigens. Both pathways were presented by the same antigens of our data, thereof 11 antigens were higher reactive in MEN-pre (RPS2, RPL13, RPL37A, RPS17, RPL23, RPS15, RPL18A, RPS21, EIF4A2, RPL8 and EIF3C), and a different set of 9 antigens (EIF4G1, RPL5, RPS11, RPL21, EIF2B4, RPL4, RPS26, EIF3L and RPS4X) were higher reactive in MEN-post.

### 2.5. Comparing Antigenic Reactivity Pathways to GBM Gene-Expression Pathways

To analyze signaling pathways enriched from gene expression data, we filtered the OncoDB gene expression data, using a cut-off of log2 fold change > 1 and FDR < 0.05, and identified 2164 genes that were overexpressed in GBM and, vice versa, 2960 genes overexpressed in normal tissue. We then performed a Reactome pathway analysis for both gene lists. Of the genes overexpressed in GBM, 119 pathways, and for genes overexpressed in normal controls, 44 pathways, were found using an FDR < 0.1 cut-off. The 25 most important pathways are shown in Table 2A,B.

Looking up these pathways (from Appendix A) in the list of pathways derived from differentially reactive antigens, the pathways are found as follows: for contrast (a), MHC class II antigen presentation and neutrophil degranulation; for contrast (b), regulation of expression of SLITs and ROBOs; for contrast (c), SRP-dependent co-translational protein, targeting to membrane and selenoamino acid metabolism, and for contrasts (d) and (e), DNA strand elongation and platelet degranulation, respectively. While a few pathways do intersect with contrasts (a–e), an impressive 17 out of 20 pathways elucidated by differential antigenic reactivity in MEN-pre vs. MEN-post overlapped with GBM-gene-expression-derived pathways (see column “*FDR-Expr*”, which depicts the FDR-values from gene-expression Reactome pathway analysis): thus, differential antigenic reactivity in MEN-pre vs. MEN-post shows a high overlap with pathways associated with genes found overexpressed in GBM vs. normal. Of the 44 pathways derived from overexpressed genes in normal vs. GBM, only 4 pathways (FDR < 0.1) intersected with the contrasts: ion homeostasis (a); interaction between L1 and Ankyrins (b); unblocking of NMDA receptors, glutamate binding and activation, and L1CAM interactions (e).

## 3. Discussion and Conclusions

The formation and presence of tumor-associated antibodies is a well-known accompaniment to cancer cell transformation and disease progression. Because antibodies have excellent molecular stability, and are potentially formed before tumor clinical manifestation, they offer potential biomarkers. In addition to their potential diagnostic or prognostic value, previous work has demonstrated the possibility of elucidating a novel omics layer of analytics and biological information from antibody profiles, using pathway analysis; therefore, we performed antibody profiling on a 16 k protein array displaying antigenic proteins derived from expression clones (covering 8173 different human transcripts), to identify TAA signatures and pathways of preoperative and postoperative glioblastomas and meningiomas vs. healthy controls.

The IgG concentrations in the sample groups were reduced in postoperative GBM patients (not significant) and in postoperative MEN patients. These differences were compensated for by using adjusted IgG concentrations on the protein arrays: thus, 110 samples from preoperative and postoperative tumors and healthy controls were probed on 16 k protein microarrays, and higher reactive DIRAGs were examined using pathway analysis in the Reactome pathway browser. In addition, findings for GBM were compared with corresponding gene expression data from the OncoDB data repository. Tumor-associated antibodies could be of diagnostic interest for minimally invasive early detection applications, and could also provide insights into tumor characteristics and pathogenesis. We found 246 DIRAGs and 375 DIRAGs in pre-surgery glioblastomas and meningiomas samples, respectively, compared to healthy controls A.

Gahoi et al. used a similar protein microarray-based approach, to analyze cerebrospinal fluid (CSF) samples from patients with low-grade glioma (LGG) and glioblastoma multiforme (GBM), and identified the antigenic response of NOL4 and KALRN in GBM, while UTP4 and CCDC28A were found as putative tumor-associated antigens in LGG: among these antigens, we only identified UTP4 in our lists of DIRAGs [18].

Syed et al. independently performed another similar study on HuProt^TM^ arrays, which presented > 17,000 proteins recombinantly expressed in S. cerevisiae. While no in-depth comparative study was performed, STUB1 and YWHAH, which were dysregulated in grade II glioma patients, were found in our study in the lists of DIRAGs: STUB1 (in contrast (e)—higher in GBM-pre compared to GBM-post, and in contrast (d)—higher in MEN-post vs. healthy) and YWHAH (in contrast (b)—higher in GBM-post vs. healthy, and in contrast (f)—higher in MEN-post compared to MEN-pre) [19]. Using the same approach to analyzing meningiomas, Gupta et al. published differentially reactive proteins IGHG4, STAT6, CRYM, CCNB1 and SELENBP1, but these were not found in our DIRAGS, although they were present on our array [20].

A study focusing on detection of the glioblastoma peptidome, for discovering novel tumor-associated antigens for immunotherapy, exploited the HLA-bound peptides from HLA-A*02(+) glioblastomas, and investigated a subset of 10 glioblastoma-associated peptide antigens in more detail: of those, 2 (derived from proteins BCAN and FABP7) were found higher reactive only in MEN-post when compared to MEN-pre [21].

In another study, Pallasch et al. used SEREX technology, and found that the antigens GLEA1, GLEA2 and PHF3 and occurrence of autoantibodies were associated significantly with prolonged survival of glioblastoma patients [22]. Interestingly, we found that PHF3 was present in our protein array, and that antigen was also higher in both glioblastoma and meningioma pre-surgery samples.

The intersection of the obtained protein lists showed 73 significant higher reactive DIRAGs (*p* < 0.05) overlapping between pre-surgery meningiomas and glioblastomas: these 73 were enriched in the pathways of *signaling by Rho GTPases, Miro GTPases and RHOBTB3* (*n* = 17) and *COPI-mediated anterograde transport* (*n* = 8), as well as some other pathways with lower antigens present (data not shown); these overlapping antigens may indicate a more general role for the affected tissue, when these are also found in other contrasts, e.g., in postoperative MEN samples.

Comparing antibody reactivity using different expression systems or sources of protein antigens may be difficult: for example, HuProt^Tm^ uses yeast, while we used *E.coli* expression clones; in our experience, using different platforms could be even more difficult if protein coupling and immobilization on different surfaces affect steric orientation, protein folding and presentation of different epitopes, leading to altered seroreactivities. These effects are particularly critical if minor differences are to be detected, as is the case with tumor-associated antibody reactivities—in contrast to the detection of seroreactivity in infectious diseases or in vaccination against, e.g., SARS-CoV-2. Overall, therefore, we rate the findings of some individual antigens in comparison to published studies as good confirmation of the relevance of our data; consequently, in order to explain the biological significance and interpretation of antibody profiling in a broader context, we would rather discuss it at the pathway level, as below.

Although the involvement and role of tumor-associated antibodies is considered a byproduct of tumor development and progression, IgG profiling could provide insights into the pathogenesis of the disease. In addition to the identification of (new) autoantigens for diagnostic applications, conceivable therapeutic targets could also be identified.

### Pathways: Discussion

Findings of the Reactome pathway analyses have been described, and also linked to gene expression data from the OncoDB, and are discussed in the following paragraphs, for the different GBM contrasts: (a) GBM-pre vs. healthy; (b) GBM-post vs. healthy; and (e) GBM-post vs. GBM-pre.

From contrast (a), GBM-pre vs. healthy, among antigens with higher reactivity in GBM *signaling by Rho GTPases* and *vesicle-mediated transport* pathways, we found the highest number, 29 and 24, of affected antigens.

**Rho GTPases** comprise 20 members, and belong to the Ras superfamily of small GTPases: this vast group of proteins has more than 150 members, and the involvement of Rho GTPases in cancer has been controversial, as the identification of the first members of this branch of the Ras superfamily of small GTPases. Initially, no direct involvement in cancer progression was observed, until numerous observations revealed dysregulation of Rho-regulated signaling pathways in cancer. Finally, point mutants in the Rho GTPases Rac1, RhoA and Cdc42 in human tumors have confirmed that Rho GTPases serve as oncogenes in several human cancer types [23]. The involvement of the Rho family of GTPases in the regulation of invasion and migration of glioblastoma cells was reviewed by Al-Koussa [24]. Alterations in antibody profiles may occur, associated with overexpression of Rho GTPases, which leads to aberrant signaling of Rho GTPases, and is commonly found in many human cancers. Similarly, any altered expression of proteins associated with these pathways could lead to increased IgG antigenic reactivity, as found.

***The vesicle-mediated transport*** pathway was found to be enriched when 24 antigens were higher reactive in GBM than in healthy IgG. In respect of that pathway, extracellular vesicles have been studied and published in the context of GBM, and are released from glioblastomas to then modulate the tumor microenvironment. The extracellular vesicles released by tumor microenvironment cells could also modulate glioblastoma cells, and GBMs utilize different communication pathways. EV-mediated communication has unique features, compared to the other communication pathways mentioned, as it allows delivery of the vesicle cargo, not only in the tumor environment—when both tumor cells and the surrounding cells can communicate via EVs—but also in remote locations. Functional aspects, such as modulation of the tumor microenvironment by GBM-derived EVs on monocytes, macrophages, microglia, T cells, endothelial cells, astrocytes and glioma stem cells, were reviewed by Matarredona and Pastor [25]. The results we found—that the proteins affecting vesicle-mediated transport are more highly reactive—are consistent with the fact that cancer cells are known to produce larger numbers of vesicles. Although the connection of the vesicle-mediated transport pathway is relevant in tumor biology, our results provide a first hint of the need to further investigate the impact of changes in seroreactivity on these proteins, and the function.

Comparing antigenic reactivities in GBM-pre to healthy sera, 10 antigens with higher reactivity in healthy sera put the **COPI-mediated anterograde transport** pathway at the forefront of the pathway analysis: this shows that this signaling pathway could be affected and disrupted, due to antigenic reactivity. Twenty-one antigens representing the **neutrophil degranulation** pathway were higher reactive in healthy individuals. When we analyzed the gene expression data set, this pathway was also enriched highly significant (*p* < 10^−6^). Sippel et al. studied immunosuppression in patients with GBM and in normal donors, and found that peripheral cellular immunosuppression in patients with GBM is associated with degranulation of neutrophils and elevated levels of circulating serum arginase I (ArgI) [26]: the authors also showed that T cell function can be restored in these individuals, by targeting ArgI, offering a potential therapeutic window to enhance antitumor immunity in affected patients; furthermore, analysis of the gene expression pathway also showed the MHC class II antigen presentation pathway significantly (*p*= 0.0025) enriched for genes overexpressed in GBM, and links to antitumor immunity.

A similar link to tumor immunity could be the *n* = 15 antigens, which are also more reactive in healthy subjects, and overrepresent the **asparagine N-linked glycosylation** pathway. As reviewed by Mereitir et al. [27], in the context of cancer, glycosylation is a tightly regulated multistep process: changes occur in cancer, and various serological tumor marker assays are based on the quantification of glycoconjugates in the serum of cancer patients, e.g., CA19-9, CA125. Functionally, glycans control or influence multiple aspects of cancer cell biology, and these biological processes underlie critical cancer hallmarks, such as invasion, angiogenesis and metastasis, involving modulation of the immune response. Conversely, changes in glycan biosynthesis can lead to the formation of immunogenic glycan neoantigens: this may reflect the situation found in our antibody profiling data, when antigenic reactivities to proteins in the asparagine N-linked glycosylation pathway are altered. Glycosylation changes in GBM are biologically highly relevant, contributing to both cancer growth and metastasis, and have also been shown to be cancer biomarkers: thus, both diagnostically and therapeutically, they are very interesting [27,28].

The ***mRNA-splicing major*** and ***axon guidance*** pathways were found to be the top pathways with higher reactive antigens in GBM-post, when compared to healthy sera (contrast (b)). **mRNA splicing major:** pre-mRNA splicing occurs within the “spliceosome”, with approximately 150 proteins present in spliceosomes, of which only a subset has been characterized. Several papers have shown that the spliceosome is affected in GBM. Fuentes-Fayos found that the dysregulation of the splicing machinery drives the development/aggressiveness of glioblastomas, when the expression of relevant spliceosome components and splicing factors has been aberrantly expressed [29]. Recently, Larionova showed that the expression level of splicing factors enables the classification of GBM patients into groups with different survival prognoses, and also reflects the phenotype of the tumor. In addition, the authors identified alternative splicing events that could regulate the GBM phenotype [30]: this was consistent with Correa, 2016, who found that GBM was associated with poor prognosis when 21 RNA-binding proteins (RBPs) that were regulators of co- and post-transcriptional events—and, in particular SNRPB, the core component of the spliceosome machinery—were overexpressed in GBM [31]. In addition, Yi et al. also described an association of genes mainly involved in the ribosome and spliceosome pathway with temozolomide resistance [32]. Another study even made a connection to extracellular vesicle-mediated transport, as described above, when it found that apoptotic GBM cell-derived EVs promote the proliferation and therapy resistance of surviving tumor cells, by secreting apoptotic extracellular vesicles (apoEVs), which are enriched with various components of spliceosomes [33]. In respect of the **axon guidance pathway**, the control of axonal growth and navigation is involved in the interaction with various dysfunctional GBM pathways that control tumor cell proliferation, migration and invasion, as well as tumor angiogenesis or immune response [34]. In this context, in 2010, Xu found the ligand-receptor system, Slit2/Robo1, which strongly influences the distribution, migration, axon guidance and branching of neuron cells. Slit2 and its transmembrane receptor, Robo1, have different distribution patterns in gliomas, and Slit2/Robo1 have tumor-suppressive effects [35]. Xue et al. analyzed signaling pathways in GBM data compared to normal brain tissue, and found overlapping signaling pathways from gene and miRNA expression, which included ion transport, positive regulation of macromolecule metabolic process, cell cycle and axon guidance as the main enriched signaling pathways [36]. Similarly, differentially expressed genes were also found, by Wang et al., that are involved in enriched signaling pathways, such as axon guidance [37].

Antigens overrepresented in ***hemostasis*** and ***G2/M transition*** pathways were found to be higher reactive in healthy sera compared to reactivities in GBM-post, for contrast (b). **Hemostasis:** brain vasculature functions are subverted during the development of brain tumors. Vascular perturbations are thought to contribute to disease progression and comorbidities, including thrombosis and hemorrhage; however, better understanding of these molecular linkages is needed, to pave the way to more effective (targeted) therapy, prophylaxis, adjunctive use of anticoagulants, and other agents able to modulate interactions between brain tumors and the coagulation system [38]. These authors have also demonstrated an association between the expression profiles of coagulation-associated genes (coagulome) in glioblastoma multiforme (GBM), and have discussed the coagulation system effectors that potentially act as targets and inducers of tumor progression [39]. The expression of the inhibitors of the coagulation and fibrinolysis systems was evaluated in gliomas of varying degrees of malignancy: expression of antigens and coagulation/fibrinolysis inhibitors in the tissues of gliomas with varying degrees of malignancy seemed to be indicative of their altered role in gliomas, going beyond that of their functions in the hemostatic system [40]: this, and other work, shows an association between cancer and hemostasis, which is also consistent with our results [40,41,42,43]. Cell cycle and **G2**/**M transition** pathways are often dysregulated in cancer, and cell cycle dysregulation is a hallmark of tumor cells [44,45,46]. Inhibition of GBM cell proliferation through G2/M cell cycle arrest is a therapeutic target [47,48], and the relevance has also been shown for glioma development, via a disturbed regulation of the G2/M phase transition, in several studies [47,48,49,50,51,52]: therefore, in our analysis, altered antibody profiles could reflect these changes.

Pathway analysis of GBM-post vs. GBM-pre—contrast (e)—generally showed a lower number of antigens enriched in pathways. The two main pathways, with 7 and 15 antigens, that were more reactive in GBM-pre were the ***platelet degranulation*** and the ***metabolism of lipids*** pathways. Any publication is currently found in PubMed when **platelet degranulation** and glioblastoma are searched. Although several publications related to cancer can be found in the literature, only one study examined the platelet proteome, and found it to be unaltered in patients with brain tumors, but impaired in lung cancer [40]: therefore, we would conclude that the association with this pathway might be a minor issue in glioblastomas. Recent literature has demonstrated the relevance, in the context of the ***metabolism of lipids*** pathway: Shakya et al. have shown a difference in gene expression in associated genes for glioblastoma stem cell niches and non-stem cell niches [53]. The role of lipids in GBM is also shown. Lipid metabolism is abnormally regulated in gliomas, and GBM tumors also accumulate more fatty acids than surrounding normal brain tissue [54], and act as energy stores [55], which can drive GBM cell proliferation [56]. Lipid metabolism has emerged as a potential therapeutic target to treat GBM and brain metastases [54,55,56,57,58,59]: see also the recent review by Kou, 2022 [60].

Antigens found enriched in meningiomas analyses have been similarly analyzed using the Reactome pathway browser, and are described in the Results section. In addition to the **COPI-mediated anterograde transport** and **asparagine N-linked glycosylation** pathways, which were also found in the GBM contrasts, the **infectious disease** and the **signaling by receptor tyrosine kinases** pathways were higher reactive in MEN-pre, when compared to healthy sera. Antigens higher reactive in healthy vs. MEN-pre were enriched for the **asparagine N-linked glycosylation** and the **metabolism of amino acids and derivatives** pathways. In the list of DIRAGs from the class comparison of higher reactive in MEN-post compared to controls, **COPI-mediated anterograde transport** and **ER-to-Golgi anterograde transport** were top of the identified pathways. Dai et al. have published KEGG pathways in meningioma from gene expression analysis, and found the AGE-RAGE signaling pathway in diabetic complications, the PI3K-Akt signaling pathway, ECM-receptor interaction and cell-adhesion molecules under the top pathways [61]. A direct comparison of the Reactome and KEGG pathways is not possible, and analysis of the gene expression data is beyond the scope of this work.

Comparing the Reactome pathways from the different contrasts in our results, we find multiple pathways in multiple contrasts, like *asparagine N-linked glycosylation, COPI-mediated anterograde transport, MHC class II antigen presentation*, *transcriptional regulation by E2F6, regulation of expression of SLITs and ROBOs*, having antigens enriched in GBM and MEN contrasts. *Selective autophagy in GBM, and SUMOylation of DNA methylation proteins* were found in several GBM contrasts, while *cap-dependent translation initiation, ER-to-Golgi anterograde transport, infectious disease metabolism of amino acids and derivatives, selenoamino acid metabolism, and SRP-dependent cotranslational protein targeting to membrane* have been present in multiple MEN contrasts: thus, when we compare pre- and post-operative vs. healthy with itself, it is not unexpected to find several pathways in different contrasts, reflecting the common etiology of the disease, but also the difference between both tumors. While a class comparison analysis to delineate differentially reactive antigens between GBM and MEN is technically feasible, and the number of DIRAGs has been indicated in Figure 3, we note that the biological difference could be better interpreted through the list of pathways presented. For GBM, we had the opportunity to compare signaling pathways identified by our antibody profiling to OncoDB signaling pathways inferred from gene expression profiling. While direct correlation analysis was not feasible in a patient- and sample-wise manner, we took the significant differentially expressed transcripts, and correlated fold change values with DIRAGs of the contrasts (a) GBM-pre vs. controls and (b) GBM-post vs. controls. Neither the entire set nor a subset of the top 25 or top 50 DIRAGS, sorted by their fold change, showed a significant correlation with gene expression data (data not shown): thus, gene expression differences may not be a direct determinant of antigenic reactivity. However, within the top 25 gene expression pathways derived from genes overexpressed in GBM, many of them are associated and linked to “immunological and inflammatory” components, as shown in Appendix A, while the transcripts overexpressed in normal tissue present pathways associated with neural function: this could represent the relevance of the immunological components, and possibly indicate the importance of the association with the antigenic profiles in GBM shown here.

Antibody profiling data using highly-multiplexed-by-array-based platforms is sparse, and a biological interpretation has been performed in very few studies. While direct comparison of study results is difficult, we found several antigens, from published work, matching our data, and made a conclusive description of antigenic reactivities in the context of expression data. The findings described have the potential to support both diagnostic and therapeutic development, particularly with regard to immuno-oncology therapy design and regimen. A broader application and analysis of antigen reactivity as a sole analysis, or in combination with other molecular analytes or omics layers, can be performed easily, and with a very small amount of 10–20 µL of serum or plasma: consequently, this technique has a significant advantage over other omics. As we have demonstrated in our work, antibody testing reflects another biological layer of disease pathology that may help improve diagnostics, reveal more biological detail and potential disease subsets, and allow future stratification and monitoring of treatment response.

## 4. Materials and Methods

### 4.1. Samples

Serum from newly diagnosed preoperative and postoperative glioblastoma (GBM) and meningioma (MEN) patients without any prior treatment or surgery were collected. The cohort was appended with age- and gender-matched serum from cancer-free controls (*n* = 48). Both meningioma and glioblastoma patients were operated on at the Vienna General Hospital (Vienna, Austria). Clinical data, sample collection and storage were previously described in detail [62]. We had planned to analyze 144 samples (48 controls, and each of the 24 glioblastoma and meningioma patients’ samples matched for pre- and post-surgery). Unfortunately, we broke 34 arrays while processing, leaving only 110 arrays to be analyzed, and so, for 110 of these 144 samples, the antibody profiles derived from 16 k protein array data consisted of samples as follows:

The GBM sample set (*n* = 28) consisted of 12 pre-surgery and 16 post surgery sera [63]. The MEN sample set (*n* = 55) consisted of 26 pre-surgery and 29 post surgery sera [62], and 27 controls. The patient *serum* samples were collected after overnight fasting on the day of surgery, and 3–4 weeks after the surgery, and were stored at −80 °C, according to the Institutional Review Board protocols approved by Medical University of Vienna. This study was carried out in accordance with the Good Scientific Practice recommendations of the ethics committee of the Medical University of Vienna (190/2011). All subjects gave written informed consent, in accordance with the Declaration of Helsinki.

### 4.2. Isolation of Immunoglobulin G

IgG from serum was purified using the Melon Gel™ IgG Spin Purification Kit (Thermo Fisher Scientific™, Vienna, Austria, Cat. No. 45,206). The serum was diluted 1:10 with kit buffer, and isolation was performed according to the manufacturer’s instructions. Concentration was determined by duplicate measurements, using A280 spectrophotometry (Epoch Take3 system). The IgG concentration was adjusted to 0.4 mg/mL, with the buffer provided in the kit, and stored at −20 °C until the slide processing.

IgG from serum was purified using the Melon Gel™ IgG Spin Purification Kit (Thermo Fisher Scientific™, Vienna, Austria, Cat. No. 45,206). The serum was diluted 1:10 with kit buffer, and isolation was performed according to the manufacturer’s instructions. Concentration was determined by duplicate measurements, using A280 spectrophotometry (Epoch Take3 system). The IgG concentration was adjusted to 0.4 mg/mL, with the buffer provided in the kit, and stored at −20 °C until the slide processing.

### 4.3. Protein Microarray Processing

AIT’s 16 k protein microarray was used for antibody profiling studies. This array comprises proteins expressed from *Escherichia coli*—cDNA expression clones and technical controls. These represent *n* = 6124 annotated human genes, and 8173 different human transcripts (and corresponding proteins expressed in one-to-several *Escherichia coli*)—cDNA expression clones derived from the UniPex expression library (provided by engine, Berlin, Germany, previously ImaGenes, previously RZPD Ressourcenzentrum für Genomforschung, Germany). Production, purification and spotting of recombinant proteins has been previously described in detail [6,7,64,65]. Briefly, concentration adjusted eluates of 6xHisTag proteins and control spots (bovine serum albumin, human serum albumin, human IgG, crude *E.coli lysate* and elution buffer) were spotted using 48-pin contact printing with a NanoPrint^TM^ LM210 device on SU8 epoxy-dip-coated slides. The slides were vacuum-sealed, and stored at 4 °C until processing. The quality control of the microarray slides included incubation with an anti-His Tag antibody, visual inspection and a qualification experiment—showing high reproducibility, as published [8]. Microarray processing was conducted as previously described, and as outlined in brief [6,7,8,65].

For slide probing, 16 k slides were equilibrated to room temperature, and blocked with DIG Easy Hyb^TM^ (Roche, Basel, Switzerland) for 30 min at RT with agitation in glass tanks, followed by three washes with 1X PBS pH 7.4 0.1% Triton X-100 (PBST; Gibco^TM^ -Thermo Fisher Scientific, Vienna, Austria, 70011044 and Merck, Vienna, Austria, X100) for 5 min each. The slides were rinsed with Milli-Q^®^ water, and spin-dried at 900 rpm for 4 min. Sample dilution of 400 µL (final concentration, 0.2 mg/mL IgG in 1X PBST 3% skimmed milk powder (Maresi Fixmilch)) was applied to each clean gasket slide (Agilent Technologies, Vienna, Austria G2534-60003, clean and dust-free), placed in clean and dust-free hybridization chambers (Agilent G253A), and closed. Air bubbles were removed, and the chambers were placed in a hybridization oven, and incubated for 4 h at RT with 12 rpm. After the incubation slides were opened, the slides were arranged in glass carriers, and washed three times in glass tanks with PBST for 5 min at RT with stirring, rinsed with Milli-Q^®^ water, and spin-dried. For IgG detection, the slides were incubated in a 1:10,000 dilution of Alexa Fluor 647 goat anti-human IgG (Life Technologies-Thermo Fisher Scientific, Vienna, Austria,, A21445; in 1X PBST 3% skimmed milk powder) for 1 h at RT with agitation in a dark chamber. Final washing comprised three washes with PBST, Milli-Q^®^ rinse and spin drying (900 rpm for 4 min).

### 4.4. Image Acquisition and Data Extraction

The spin-dried slides were sorted into the slide carriers, and scanned with a Tecan LS 200 Microarray scanner. The TIFF images were loaded in GenePix Pro 6.0, the .gal file was manually aligned, and spots of insufficient quality were flagged. Data were extracted as raw .gpr files, and were further processed in RStudio [66] and BRB ArrayTools [67].

### 4.5. Preprocessing and Differential Reactivity Analysis

All data preprocessing and differential reactivity analysis steps were performed using RStudio (R version R 4.0.4 and R 4.1.3). Raw fluorescence intensities were loaded into the RStudio environment via the “read.maimages” functions of the limma package [68], and median fluorescence values of the spots were corrected for the local background. Flagged features were removed and subjected to log2 transformation; features were missing in more than 50% of the samples removed, and missing features were imputed via knn imputation [69]. Preprocessed (raw) intensities were inspected via boxplots and PCA, which indicated effects associated with the experimental runs: hence, the dataset was subjected to ComBat normalization [70], with run as the batch variable. ComBat-normalized data was subsequently quantile-normalized, and replicate spots were averaged.

Differentially reactive antigens (DIRAGs) were elucidated via t-tests (genefilter package [71]) with a cut-off of *p* < 0.05, and a fold change cut-off of FC = 1.25. The results lists were sorted and filtered for (significant) DIRAGs higher reactive in the respective case group, and were used as subsets for paired analysis of pre- and post-surgery samples. The intersection of higher reactive DIRAGs was conducted via the JVenn tool [72]. After data preprocessing and normalization, *n* = 8254 features remained for t-tests (genefilter package). The DIRAGs were sorted and filtered, for their respective *p*-value (*p* < 0.05) and fold-change antigenic reactivities between classes.

### 4.6. Reactome Pathway

Differential reactivity analysis result lists were sorted and filtered by fold change and proteins with higher reactivity in each case group, and were analyzed via ReactomePA—an R package for Reactome pathway analysis [73], using Reactome version 79 [13]; the full 16 k protein list was used as background (human organism, *p* < 0.05, BH correction method), and the top 10 metabolic pathways were extracted as .txt files.

### 4.7. Gene Expression OncoDB Pathway Intersection with Antigenic Pathways

OncoDB (http://oncodb.org/index.html accessed on 30 August 2022 [14]) is an online database resource for researchers to explore abnormal patterns in gene expression and viral infection that are correlated to clinical features in cancer. The patient samples we used are not part of the OncoDB. We accessed the online database in June 2022, and downloaded a differential expression data set for GBM, including Gene symbol, median log2 gene expression, MFI values of cancer and normal samples, log2 fold change, single gene *p*-values and FDR-adjusted *p*-values. Expression data were compared with antibody reactivities based on gene symbols and pathways using gene-subsets selected at a cut-off of log2 fold change >1 and FDR < 0.05, as given in the results text.

## Figures and Tables

**Figure 1 ijms-24-01411-f001:**
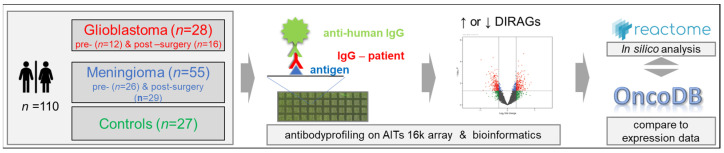
Schematic representation of the study design. The Medical University of Vienna provided 110 serum samples from preoperative and postoperative glioblastomas, meningiomas and matching cancer-free controls. IgG isolated from the samples was probed on AIT’s 16 k protein microarray, to elucidate differentially reactive antigens (DIRAGs) by class comparison of sera from tumor patients with controls, and from preoperative and postoperative samples. Lists of statistically significant antigens were then subjected to a Reactome pathway analysis [13], and were compared to glioblastoma gene expression data from OncoDB (http://oncodb.org/ accessed on 30 August 2022; [14]).

**Figure 2 ijms-24-01411-f002:**
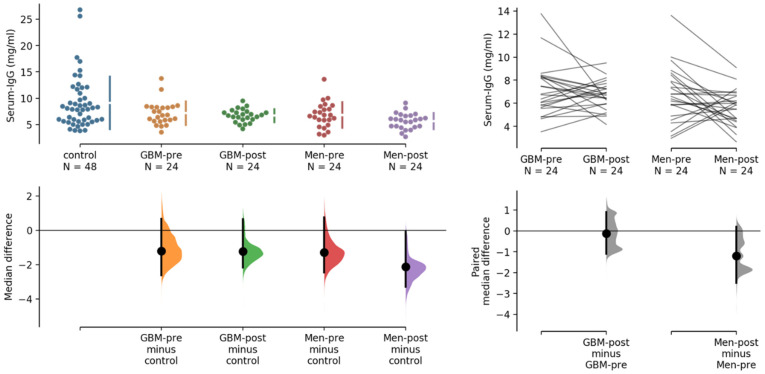
Comparative analysis of serum IgG concentrations in 144 samples, using healthy control sera as common control (left), and pairwise analysis of GBM and MEN samples before and after surgery. The serum IgG concentrations are indicated on the y-axes in the upper panels, the mean differences in the lower panels. For 110 of these samples, antibody profiling on 16 k protein array data (as shown in Figure 1) were available.

**Figure 3 ijms-24-01411-f003:**
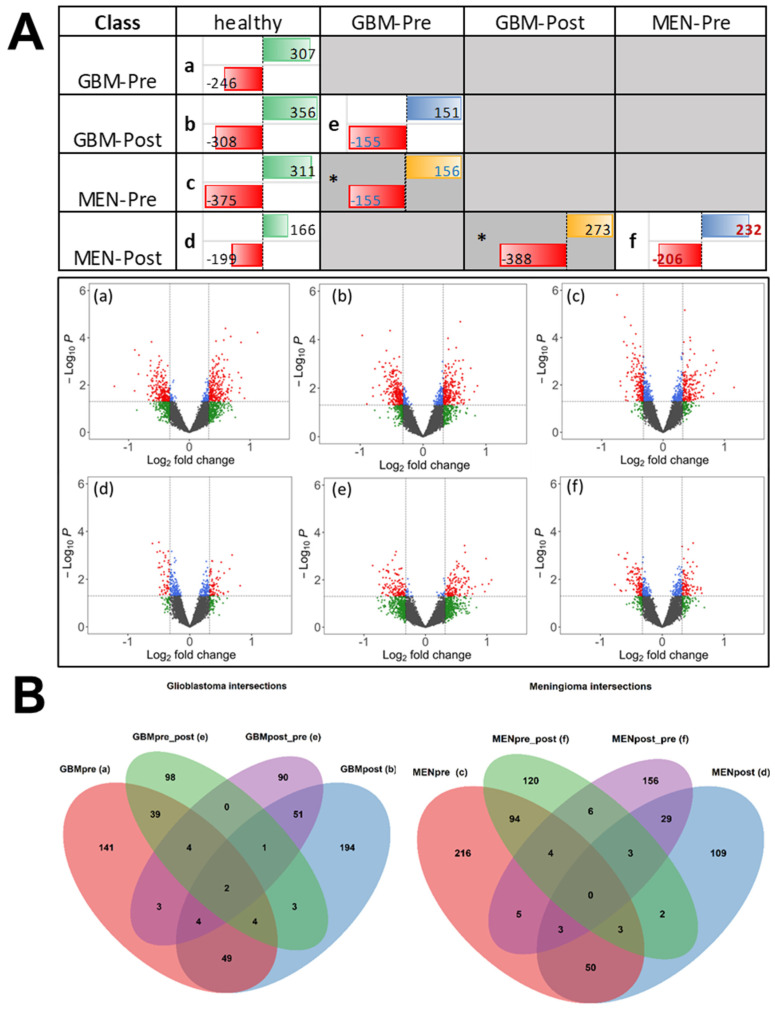
(**A**) The different contrasts (comparison of two different biological classes) (a–f) are shown, and the numbers of significantly differentially reactive antigens (DIRAGs) are indicated. Data from contrasts marked with an * asterisk are not considered, but are shown for completeness. Numbers in red cells (left side in table cells) are higher reactive in the classes in the left column “class” of the table. Numbers in green, blue, and yellow fields (right side, in table fields) are higher reactive in the classes in the first row “class” of the table. GBM: glioblastoma; MEN: meningioma; Pre: preoperative-derived serum samples; Post postoperative-derived serum samples. Volcano plots of contrasts (a–f) are shown. Log2 fold change (x-axis; cut-off fold change FC = 1.25) and *p*-values cut-off (y-axis: negative log10 of *p* = 0.05) are shown as dashed lines. Number of significant features (*p* < 0.05), as indicated in the upper crosstable, are marked as red and blue dots. Plots were generated with the EnhancedVolcano R package [17]. (**B**) Venn diagrams showing the overlap of DIRAGS for the specific contrasts of GBM (left) and MEN (right), compared to healthy controls. Overlap of GBM contrasts in left Venn diagram: 246 DIRAGS more reactive in GBM-pre, compared to controls, from contrast (a) is indicated in the red ellipse; 151 DIRAGS higher reactive in GBM-pre, compared to GBM-post, from contrast (e) is indicated in the green ellipse; in purple, 155 higher reactive in GBM-post, compared to GBM-pre, of contrast (e); and, in blue, 308 DIRAGS higher reactive in GBM-post, compared to controls, derived from contrast (b). Overlap of MEN contrasts in the right Venn diagram: 375 DIRAGS higher reactive in MEN-pre, compared to controls, from contrast (c) is indicated in the red ellipse; 232 DIRAGS higher reactive in MEN-pre, compared to MEN-post, from contrast (f) is indicated in the green ellipse; in purple, 206 higher reactive in MEN-post, compared to MEN-pre, from contrast (f); and, in blue, 199 DIRAGS higher reactive in MEN-post vs. controls, from contrast (d).

**Table 1 ijms-24-01411-t001:** Toptables of DIRAGS in (A) glioblastomas (contrasts (a,b,e)) and (B) meningiomas (contrasts (c,d,f)). Antigens are sorted by the fold change reactivity between classes, and the top 10 antigens of both classes for each contrast are indicated with t-statistics, *p*-values and AUC values. Gene symbols are indicated. Complete lists of DIRAGs are given in Appendix A.

A Contrast	Higher in	t-Statistic	Fold Change	*p*-Value	AUC	Gene	B Contrast	Higher in	t-Statistic	Fold Change	*p*-Value	AUC	Gene
a-GBM-PRE vs. control	gbm-pre	−2.615	−1.234	0.011	0.702	VAT1L	c-MEN-PRE vs. control	men-pre	−5.229	−0.749	0.000	0.816	SART1
		−2.441	−0.901	0.018	0.726	RNF213			−2.775	−0.698	0.007	0.639	TLE2
		−3.816	−0.897	0.000	0.741	ARFGAP2			−2.617	−0.692	0.011	0.692	MORF4L1
		−3.049	−0.837	0.003	0.687	KAT14			−4.001	−0.639	0.000	0.729	DDX11
		−3.660	−0.829	0.001	0.771	ARFGAP2			−4.666	−0.626	0.000	0.782	FAM209B
		−3.002	−0.691	0.004	0.753	BTBD7			−2.052	−0.608	0.044	0.631	CCNL2
		−2.302	−0.689	0.025	0.721	USP54			−2.191	−0.606	0.032	0.604	MED4
		−2.623	−0.668	0.011	0.687	RPL37A			−2.395	−0.596	0.019	0.698	DDX18
		−2.079	−0.668	0.042	0.725	CCT6A			−3.898	−0.592	0.000	0.764	TTC3
		−3.328	−0.640	0.002	0.807	PCDHB14			−2.946	−0.589	0.004	0.689	MED7
	control	2.443	0.688	0.018	0.719	MYBBP1A		control	2.612	0.662	0.011	0.677	NFIX
		2.569	0.700	0.013	0.752	GCN1			3.053	0.692	0.003	0.695	MAP1LC3B
		2.826	0.704	0.006	0.798	SNX15			3.310	0.700	0.001	0.684	DMPK
		3.099	0.808	0.003	0.844	ALG3			2.975	0.766	0.004	0.695	MAZ
		2.371	0.823	0.021	0.736	NFIX			3.111	0.784	0.003	0.720	ALAD
		2.542	0.833	0.014	0.752	CUL9			3.750	0.817	0.000	0.753	OTUD1
		4.039	0.851	0.000	0.781	EIF4EBP1			2.324	0.822	0.023	0.697	TACC2
		2.927	0.857	0.005	0.818	GIPC1			3.148	0.836	0.002	0.706	RALGDS
		2.141	0.909	0.036	0.790	ZNF232			3.386	0.871	0.001	0.710	RALGDS
		4.324	1.123	0.000	0.819	TELO2			2.552	1.157	0.013	0.697	TTLL12
b-GBM-POST vs. control	gbm-post	−4.271	−0.968	0.000	0.757	DBN1	d-MEN-POST vs. control	men-post	−3.775	−0.603	0.000	0.723	INPP5B
		−2.045	−0.894	0.045	0.707	VAT1L			−3.526	−0.526	0.001	0.704	PSMB5
		−2.670	−0.798	0.010	0.754	CCNL2			−3.040	−0.506	0.003	0.685	EEF1A1
		−2.359	−0.797	0.021	0.697	CEP57			−3.809	−0.499	0.000	0.743	LMF2
		−3.301	−0.766	0.002	0.756	ZNF341			−3.188	−0.492	0.002	0.691	ISCU
		−2.757	−0.696	0.008	0.754	STK11IP			−2.340	−0.486	0.022	0.679	ERP29
		−3.512	−0.695	0.001	0.722	SNAP47			−2.047	−0.483	0.044	0.648	BCL9
		−2.413	−0.669	0.019	0.699	UCHL1			−3.553	−0.482	0.001	0.712	ARFGAP2
		−3.359	−0.650	0.001	0.763	PER1			−3.044	−0.482	0.003	0.664	SLC20A2
		−2.195	−0.635	0.032	0.673	EDC4			−2.802	−0.461	0.006	0.689	IGHA1
	control	3.568	0.636	0.001	0.768	CUL7		control	2.688	0.487	0.009	0.665	ALG3
		2.045	0.641	0.045	0.682	MTMR14			2.138	0.498	0.036	0.616	NFIX
		2.097	0.651	0.040	0.666	ALKBH5			3.213	0.508	0.002	0.728	NPC2
		2.208	0.654	0.031	0.625	PLOD1			2.112	0.542	0.038	0.625	PRKRA
		2.567	0.654	0.013	0.724	BMS1			2.066	0.548	0.042	0.635	GMIP
		2.709	0.665	0.009	0.717	RC3H2			2.856	0.556	0.006	0.700	ALG12
		2.220	0.736	0.030	0.652	RALGDS			2.168	0.619	0.033	0.647	STAT5A
		3.317	0.758	0.002	0.793	ALG3			2.997	0.644	0.004	0.709	PPP2R1A
		2.528	0.814	0.014	0.655	KIFAP3			3.435	0.684	0.001	0.708	OTUD1
		2.737	0.864	0.008	0.718	NFIX			2.402	0.815	0.019	0.678	TACC2
e-GBM-PRE vs. GBM-POST	gbm-pre	−3.350	−0.858	0.002	0.859	EIF3G	e-MEN-PRE vs. MEN-POST	men-pre	−2.473	−0.766	0.017	0.647	CCNL2
		−2.585	−0.751	0.016	0.771	POLD2			−3.509	−0.705	0.001	0.762	DDX11
		−3.112	−0.749	0.004	0.792	GPATCH1			−2.252	−0.678	0.028	0.664	LCK
		−2.199	−0.706	0.037	0.766	GTF2IP4			−2.215	−0.659	0.031	0.627	TLE2
		−2.229	−0.693	0.035	0.682	TPP2			−3.383	−0.611	0.001	0.767	RPS2
		−2.465	−0.690	0.021	0.734	ARHGAP33			−2.240	−0.586	0.029	0.745	CDC37
		−2.873	−0.690	0.008	0.797	NAA20			−3.380	−0.585	0.001	0.781	GSTK1
		−3.203	−0.685	0.004	0.891	PLEKHA5			−2.652	−0.569	0.011	0.713	PTK7
		−2.404	−0.677	0.024	0.755	TAX1BP3			−3.108	−0.565	0.003	0.733	N4BP3
		−2.437	−0.663	0.022	0.813	GOLGB1			−3.567	−0.564	0.001	0.759	KIF2C
	gbm-post	2.801	0.705	0.009	0.792	GIPC1		men-post	2.237	0.491	0.030	0.685	MSH2
		2.462	0.712	0.021	0.734	SYNE2			3.271	0.498	0.002	0.753	RABGGTB
		2.411	0.753	0.023	0.820	SLC7A5			3.871	0.500	0.000	0.776	AZI2
		2.170	0.767	0.039	0.724	SAP30BP			2.891	0.503	0.006	0.696	PSMB5
		2.492	0.809	0.019	0.802	EIF4EBP1			2.130	0.530	0.038	0.639	RPS26
		2.584	0.925	0.016	0.771	CCNL2			2.266	0.558	0.028	0.651	MAP1LC3B
		2.275	0.967	0.031	0.771	BAIAP2			3.127	0.559	0.003	0.749	NUCB1
		3.615	0.986	0.001	0.870	LAMTOR2			2.291	0.570	0.026	0.687	RALGDS
		2.611	1.030	0.015	0.708	TELO2			2.278	0.622	0.027	0.634	DMPK
		2.775	1.074	0.010	0.807	IGKC			2.349	0.625	0.023	0.678	ALAD

**Table 2 ijms-24-01411-t002:** (**A**) Upper table: top 25 pathways derived from the reactome analysis of genes overexpressed in GBM, using the OncoDB data set. (**B**) Bottom table: likewise for genes overexpressed in normal tissue.

(A) TOP25-Higher in GBM	Entities	Entities	Entities	Entities	Entities	Reactions	Reactions	Reactions
Pathway Name	Found	Total	Ratio	* p-Value *	FDR	Found	Total	Ratio
Endosomal/Vacuolar pathway	77	82	0.005	1.11 × 10^−16^	2.42 × 10^−14^	*4*	*4*	*0*
SRP-dependent cotranslational protein targeting to membrane	68	119	0.008	1.11 × 10^−16^	2.42 × 10^−14^	*5*	*5*	*0*
ER-Phagosome pathway	97	173	0.011	1.11 × 10^−16^	2.42 × 10^−14^	*10*	*10*	*0.001*
Antigen processing-Cross presentation	107	195	0.013	1.11 × 10^−16^	2.42 × 10^−14^	*22*	*23*	*0.002*
Antigen Presentation: Folding, assembly and peptide loading of class I MHC	77	108	0.007	1.11 × 10^−16^	2.42 × 10^−14^	*15*	*16*	*0.001*
Interferon Signaling	180	395	0.026	1.11 × 10^−16^	2.42 × 10^−14^	*33*	*71*	*0.005*
Cytokine Signaling in Immune system	338	1094	0.072	1.11 × 10^−16^	2.42 × 10^−14^	*244*	*710*	*0.051*
Interferon gamma signaling	156	250	0.017	1.11 × 10^−16^	2.42 × 10^−14^	*5*	*16*	*0.001*
Interferon alpha/beta signaling	104	190	0.013	1.11 × 10^−16^	2.42 × 10^−14^	*7*	*24*	*0.002*
SARS-CoV-2-host interactions	124	314	0.021	2.22 × 10^−16^	4.35 × 10^−14^	*20*	*67*	*0.005*
Peptide chain elongation	59	97	0.006	7.77 × 10^−16^	1.38 × 10^−13^	*4*	*5*	*0*
Nonsense Mediated Decay (NMD) independent of the Exon Junction Complex (EJC)	59	101	0.007	4.33 × 10^−15^	7.06 × 10^−13^	*1*	*1*	*0*
Eukaryotic Translation Elongation	59	102	0.007	6.55 × 10^−15^	9.89 × 10^−13^	*7*	*9*	*0.001*
Immunoregulatory interactions between a Lymphoid and a non-Lymphoid cell	119	316	0.021	2.04 × 10^−14^	2.86 × 10^−12^	*39*	*44*	*0.003*
Formation of a pool of free 40S subunits	59	106	0.007	3.26 × 10^−14^	4.24 × 10^−12^	*2*	*2*	*0*
Eukaryotic Translation Termination	58	106	0.007	1.07 × 10^−13^	1.31 × 10^−11^	*3*	*5*	*0*
SARS-CoV-2 activates/modulates innate and adaptive immune responses	93	226	0.015	1.25 × 10^−13^	1.44 × 10^−11^	*17*	*47*	*0.003*
Response of EIF2AK4 (GCN2) to amino acid deficiency	60	115	0.008	2.90 × 10^−13^	3.16 × 10^−11^	*6*	*16*	*0.001*
Selenocysteine synthesis	58	112	0.007	9.59 × 10^−13^	9.88 × 10^−11^	*2*	*7*	*0.001*
L13a-mediated translational silencing of Ceruloplasmin expression	60	120	0.008	1.62 × 10^−12^	1.58 × 10^−10^	*3*	*3*	*0*
** (B) TOP25-Higher in NORMAL **	** Entities **	** Entities **	** Entities **	** Entities **	** Entities **	** Reactions **	** Reactions **	** Reactions **
** Pathway Name **	** Found **	** Total **	** Ratio **	** * p * ** ** -Value **	** FDR **	** Found **	** Total **	** Ratio **
Neuronal System	238	489	0.032	1.11 × 10^−16^	1.03 × 10^−13^	*182*	*216*	*0.016*
Transmission across Chemical Synapses	155	343	0.023	1.11 × 10^−16^	1.03 × 10^−13^	*136*	*163*	*0.012*
Neurotransmitter receptors and postsynaptic signal transmission	112	232	0.015	4.44 × 10^−15^	2.75 × 10^−12^	*105*	*109*	*0.008*
Protein-protein interactions at synapses	56	93	0.006	1.27 × 10^−11^	5.87 × 10^−09^	*32*	*33*	*0.002*
Neurexins and neuroligins	40	60	0.004	6.14 × 10^−10^	2.28 × 10^−07^	*19*	*19*	*0.001*
Potassium Channels	56	107	0.007	1.90 × 10^−09^	5.88 × 10^−07^	*14*	*19*	*0.001*
Activation of NMDA receptors and postsynaptic events	55	113	0.007	2.87 × 10^−08^	7.62 × 10^−06^	*71*	*71*	*0.005*
Post NMDA receptor activation events	48	96	0.006	9.80 × 10^−08^	2.27 × 10^−05^	*39*	*39*	*0.003*
Trafficking of AMPA receptors	25	37	0.002	7.23 × 10^−07^	1.49 × 10^−04^	*4*	*4*	*0*
Glutamate binding, activation of AMPA receptors and synaptic plasticity	25	39	0.003	1.81 × 10^−06^	3.35 × 10^−04^	*9*	*9*	*0.001*
Cardiac conduction	57	138	0.009	2.74 × 10^−06^	4.64 × 10^−04^	*24*	*27*	*0.002*
Voltage gated Potassium channels	26	44	0.003	4.78 × 10^−06^	7.36 × 10^−04^	*1*	*1*	*0*
Long-term potentiation	21	31	0.002	5.15 × 10^−06^	7.37 × 10^−04^	*7*	*7*	*0.001*
Unblocking of NMDA receptors, glutamate binding and activation	18	27	0.002	2.88 × 10^−05^	3.80 × 10^−03^	*5*	*5*	*0*
CREB1 phosphorylation through NMDA receptor-mediated activation of RAS signaling	22	39	0.003	4.83 × 10^−05^	5.94 × 10^−03^	*7*	*7*	*0.001*
LGI-ADAM interactions	12	14	0.001	6.08 × 10^−05^	6.65 × 10^−03^	*5*	*5*	*0*
CaM pathway	23	43	0.003	7.23 × 10^−05^	6.65 × 10^−03^	*23*	*24*	*0.002*
Calmodulin induced events	23	43	0.003	7.23 × 10^−05^	6.65 × 10^−03^	*22*	*23*	*0.002*
GABA receptor activation	31	68	0.005	8.74 × 10^−05^	7.47 × 10^−03^	*11*	*12*	*0.001*
Negative regulation of NMDA receptor-mediated neuronal transmission	17	27	0.002	9.34 × 10^−05^	7.47 × 10^−03^	*4*	*4*	*0*

## Data Availability

Data are available upon request.

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
