# Peer review of "Antibody Profiling and In Silico Functional Analysis of Differentially Reactive Antibody Signatures of Glioblastomas and Meningiomas"

_ijms, 2023, doi:10.3390/ijms24021411_

Round 1
Reviewer 1 Report
This manuscript conducts IgG antibody profiling on sera from glioblastoma and meningioma patients and healthy controls using a high-density protein array. Differentially reactive antigens were then subsequently compared to (public?) gene-expression data (OncoDB) and Reactome pathways.
There are several major and minor concerns that I would like to see addressed, which are detailed below.
Major concerns:
1) The manuscript length (27 pages) is typical of a thesis, rather than a research article, and the results (16 pages) and discussion (5 pages) sections require substantial shortening in accordance with the typical research article format.
2) Detailed information pertaining to patient cohorts (both in-house and publicly accessed) and sample time points is lacking and requires inclusion.
3) Figure 3 is repeatedly incorrectly referred to as Figure 1 or 2 throughout the results, and requires correction.
4) Supplementary Tables 1 and 2 are not provided (only the legends) and must be included.
Minor concerns:
1) Title: please revise to clarify that in silico analysis was performed on the identified cognate DIRAGs, not the autoantibody signatures, or consider leaving this out.
2) Abstract: Please include patient and sample numbers, and clarify that controls are healthy donors. Is the OncoDB-derived gene expression using an alternate public GBM dataset, with no patient overlap (this is unclear)?
3) Introduction: Are meningiomas considered benign (non-cancerous), and if so why have these been included (are these intended as a benign brain-specific control?)? Antibodies against cognate tumor antigens are not formed, but rather produced by anybody-securing B cells, please revise terminology throughout the manuscript. Similarly, tumor-specific antibodies cannot be autoantibodies, the latter only applies to autoantigens. Page 2, line 45 refers to previous work, but does not include a reference. GBM and MEN abbreviations are not consistently used in the manuscript.
4) Material and Methods - Are pre- and post-surgical samples matched (same patient)? Based on the blood sample numbers, there are more post-surgery samples, so this can't be the case entirely. How many total patient numbers do these blood samples equate to? What are the patient's characteristics? How long after surgery is the post-surgical blood sample, and was surgery with curative intent? How were controls matched? Include the ethical approval reference number for this study. It sounds like the AIT array includes genes and proteins, which is unlikely (re-word). Confirm that a 1:10,000 dilution of detection antibody was used.
5) Results: As mentioned above, this section requires substantial shortening and summarizing especially section 2.4. Please include a patient characteristics table. It is unclear if pre- and post-surgical blood samples are matched, which will affect the interpretation of results. Figure 2 should be labeled with a, b, c and d, and the legend revised accordingly. Figure 2 indicates n=24 for GBM-pre and post, and for MEN-pre and post, but this does not match the sample numbers provided earlier, please clarify. It is very hard to follow the analysis of the results indicated only by contrast a) to f). Figures should be placed immediately after the paragraph they are mentioned. There should be caution when reporting AUC values with such small patient numbers (n=12 for GBM-pre), and this should be mentioned in the text. Figure 3 is repeatedly incorrectly referred to as Figure 1 or 2 on page 4 (lines 147, 160, 161), page 8 (lines 193, 194, 200, 208), and page 9 (line 247). Tables 1, 2, and 3 should either be shortened/summarised or moved to supplementary materials. The axes legends of the volcano plots in figure 3A are not legible. On page 8, line 199, should this be 49 instead of 45? Does figure 4 add anything to the Venn diagrams already provided in figure 3B (if not, please remove it)? On page 9, line 252, the lowest adj p-value I see is 0.11, please revise the range. Is the OncoDB gene expression cohort data a different GBM cohort (n=?, include details of cohort)? Why would there be any interest in genes overexpressed in healthy controls (the same question applies to DIRAGs higher in controls)? Include the adj p-values for the data indicated in table 3.
6) Discussion: As mentioned above, this section requires substantial shortening. Once again, patient numbers, the pairing of pre-and post-surgical samples, and overlap with the OncoDB cohort are unclear and should be clearly stated. If surgery was curative, and the post-surgery blood sample was several months after surgery (after the maximum time of circulating antibody half-life), would you expect to see any antibodies at all? How reproducible is the array data using the AIT protein microarray (were any samples run multiple times and assessed for overlap?)? Are the immobilized proteins on the AIT protein microarrays full-length and correctly folded?
7) References: Please edit the reference format to list all authors, rather than et. al.
Reviewer 2 Report
First of all I would like to thank the authors for the extensive study of the antigens and genes involved in the GBM and MEN. The presented work is descriptive and provides the major antigens which can be used as target for disease pathogenesis and drug discovery researches.
I would like to recommend few modifications in the paper before its publication:
1. Providing more information about the autoantibodies and its relevance would help readers to get better picture of why the study is done.
2. Some images appears to be redundant which can be improved for better picturization and summarized information.
3. Section 2.1: (Line 96) Please explain the relevance or rationale for comparison of serum IgG concentrations. Is the concentration change result of disease? Also, it would be interesting to know if there is any reason behind the change in concentration.
4. Line 134: Please revisit the line and complete the sentence.
5. Table 1: Please revisit the table legends since if fold change and T-values needs to be changed. Also please add one more column in table to clearly mention the what comparison was done on each for better visualization. Also, Supplementary table contains just genes for 10 antigens. Please verify that the table is complete or remove this statement. Since it says complete only for review process.
6. Line 230: supplementary table does not contain full list, please rewrite the statement.
7. Fig. 4: Isn't this information same as venn diagram? If so, only one image can be used, this one or venn diagram in figure 3b.
8. Line 701: Reference 61 Does not include the sample collection for serum. Please remove it form the reference section.
9. Please include the section which explains the cutoff for less expressed antigens were cutoff for the result along with the fold change.
